# De-Escalating the Management of In Situ and Invasive Breast Cancer

**DOI:** 10.3390/cancers14194545

**Published:** 2022-09-20

**Authors:** Fernando A. Angarita, Robert Brumer, Matthew Castelo, Nestor F. Esnaola, Stephen B. Edge, Kazuaki Takabe

**Affiliations:** 1Division of Surgical Oncology and Gastrointestinal Surgery, Department of Surgery, Houston Methodist Hospital, Houston, TX 77030, USA; 2Department of Surgery, Houston Methodist Hospital, Houston, TX 77030, USA; 3Division of General Surgery, Department of Surgery, University of Toronto, Toronto, ON M5T 2S8, Canada; 4Breast Surgery, Department of Surgical Oncology, Roswell Park Comprehensive Cancer Center, Buffalo, NY 14203, USA; 5Department of Surgery, Jacobs School of Medicine and Biomedical Sciences, State University of New York, Buffalo, NY 12246, USA; 6Department of Gastroenterological Surgery, Graduate School of Medicine, Yokohama City University, Yokohama 236-0004, Japan; 7Department of Breast Surgery, Fukushima Medical University, Fukushima 960-1295, Japan; 8Department of Breast Surgery and Oncology, Tokyo Medical University, Tokyo 160-0023, Japan

**Keywords:** de-escalation, ductal carcinoma in situ, breast cancer, breast surgery, neoadjuvant chemotherapy

## Abstract

**Simple Summary:**

De-escalation of breast cancer treatment reduces morbidity and toxicity for patients. De-escalation is safe if cancer outcomes, such as recurrence and survival, remain unaffected compared to more radical regimens. This review provides an overview on treatment de-escalation for ductal carcinoma in situ (DCIS), local treatment of breast cancer, and surgery after neoadjuvant systemic therapy. Improvements in understanding the natural history and biology of breast cancer, imaging modalities, and adjuvant treatments have facilitated de-escalation of treatment over time.

**Abstract:**

It is necessary to identify appropriate areas of de-escalation in breast cancer treatment to minimize morbidity and maximize patients’ quality of life. Less radical treatment modalities, or even no treatment, have been reconsidered if they offer the same oncologic outcomes as standard therapies. Identifying which patients benefit from de-escalation requires particular care, as standard therapies will continue to offer adequate cancer outcomes. We provide an overview of the literature on the de-escalation of treatment of ductal carcinoma in situ (DCIS), local treatment of breast cancer, and surgery after neoadjuvant systemic therapy. De-escalation of breast cancer treatment is a key area of investigation that will continue to remain a priority. Improvements in understanding the natural history and biology of breast cancer, imaging modalities, and adjuvant treatments will expand this even further. Future efforts will continue to challenge us to consider the true role of various treatment modalities.

## 1. Introduction

Modern breast cancer treatment has evolved into a complex multi-modality process aimed at eradicating the patient’s current tumor burden while also minimizing future risk of recurrence. Over 90% of women with newly diagnosed breast cancer are projected to survive at least five years [1], therefore, interest in de-escalating treatments to decrease morbidity and preserve quality of life has grown. Improved understanding of breast cancer biology, advancements in technology, and growing interest in patient-centered outcomes have further spurred these endeavors. Although the greater body of research on de-escalation of breast cancer treatment has occurred during the last three decades, efforts started early on. As breast cancer treatment becomes personalized, de-escalation studies are providing data on what can be safely omitted in each patient. The management of patients with ductal carcinoma in situ (DCIS) and invasive breast cancer have both undergone scrutiny amidst concerns of overtreatment [2,3,4,5]. Here we provide an overview of the literature on de-escalation of treatment of ductal carcinoma in situ (DCIS), local treatment of breast cancer, and surgery after neoadjuvant systemic therapy

## 2. De-Escalation of DCIS Treatment

Currently, DCIS is treated because of the inability to accurately predict who will progress and develop invasive breast cancer. Standard treatment includes surgical resection and, potentially, local radiation therapy (RT), if the patient undergoes breast conservation to minimize the risk of local recurrence. Adjuvant endocrine therapy may be recommended, depending on the hormone-receptor status of the tumor for those treated with breast conservation, to further reduce the risk of local recurrence. The long-term survival from DCIS is robust irrespective of the type of surgery or the use of radiation with breast conservation, with 10- and 20-year cause-specific survival rates of 98.9% and 96.7%, respectively [6]. However, local recurrence after breast conservation occurs for DCIS and invasive carcinoma in equal proportions [7]. Post-lumpectomy adjuvant RT reduces the risk of ipsilateral in situ and invasive breast recurrence by half, but does not impact survival, as demonstrated by four randomized trials [8,9,10,11]. 

De-escalation of DCIS treatment has attracted attention for several reasons. DCIS has low breast cancer-specific mortality in patients treated with standard modalities, or, potentially, with no treatment. In a Surveillance, Epidemiology, and End Results (SEER) study of 108,196 patients with DCIS who underwent standard treatment, the 10- and 20-year breast cancer-specific mortality was 1.1 and 3.3%, respectively [6]. Further, autopsy studies report that among women who died of non-breast cancer causes, up to 39% had undetected DCIS, suggesting that for some women, DCIS remains indolent and will never develop into a clinically significant disease [12]. Left untreated, 10.5% to 50% of patients with DCIS may progress to invasive carcinoma [13,14,15,16]. The indolent and potentially sub-clinical nature of DCIS is further suggested by the finding that the rate of DCIS increased dramatically with the use of population screening mammography. If DCIS all progressed to invasive cancer, it would have been expected that widespread use of mammography would then result in a corresponding decrease in the rate of invasive cancer, but this did not occur [17]. Therefore, DCIS may not be an obligate precursor of invasive carcinoma [18,19,20,21]. 

Indeed, locoregional treatment may not be necessary, as some cases of estrogen receptor (ER)-positive DCIS can resolve with primary endocrine therapy (PET) [22,23]. The Cancer and Leukemia Group B (CALGB) 40903 study, a phase II single arm study of preoperative letrozole for ER-positive DCIS in postmenopausal women, reported that 15% of patients had no residual disease at the time of surgery [22]. Finally, DCIS treatments, including RT and endocrine therapy, do not improve survival and were associated with potential side effects [24,25,26]. Current research efforts aim to personalize the treatment of DCIS by de-escalating unnecessary treatment for certain types of tumors. 

A key focus of the move to de-escalate surgical treatment of DCIS is the identification of women who require no surgery after diagnosis by core needle biopsy. In lieu of surgery, patients could undergo active surveillance, with or without endocrine therapy. Four phase III prospective trials are studying active image-based surveillance for patients with low-risk DCIS: COMET (Comparing an Operation to Monitoring, With or Without Endocrine Therapy) in the US, LORD (Low Risk DCIS) in Europe, LORETTA (Low-Risk DCIS with Endocrine Therapy Alone-Tamoxifen) in Japan, and LORIS (Surgery Versus Active Monitoring for LowRisk DCIS) in the United Kingdom [27,28,29,30] (Table 1). The primary endpoint of these trials is development of ipsilateral invasive cancer. Unfortunately, because of low accrual, both the LORIS and LORD trials converted to registry trials. 

De-escalation of post-lumpectomy RT for DCIS has focused on decreasing the duration of radiation, the extent of the breast to be radiated, and even omitting radiation altogether. Hypofractionated RT, whereby the same biologic dose of whole breast radiation is given in fewer fractions, has been evaluated for the treatment of DCIS. The Danish Breast Cancer Group (DBCG) HYPO Trial, a randomized phase III trial of hypofractionation versus standard fractionated RT, included 264 patients with DCIS [31]. The 9-year incidence of locoregional recurrence was 4.9% in the 50Gy control arm and 6.5% in the experimental arm [hazard ratio (HR) 1.40, 95% confidence interval (CI) 0.49–4.05]. The Breast International Group (BIG) 3-07/Trans-Tasman Radiation Oncology Group (TROG) 07.01, a four-arm randomized trial, is evaluating post-lumpectomy hypofractionated RT versus standard RT, with or without boost in patients with high-risk DCIS [32]. 

Accelerated partial breast irradiation (APBI) is an option for certain patients with low-risk DCIS to limit the extent of breast treated. Select patients with DCIS may be considered suitable for APBI if they meet the low-risk criteria established by the Radiation Therapy Oncology Group (RTOG) 9804 trial, including screen-detected DCIS, nuclear grade 1 or 2, tumor ≤2.5 cm, and surgical resection with margins negative at > 3mm [33]. Both the National Comprehensive Cancer Network (NCCN) and American Society for Radiation Oncology (ASTRO) guidelines accept APBI in patients with low-risk DCIS who are ≥50 years old and BRCA mutation negative [34,35]. 

Local recurrence can occur after resection of any DCIS. Radiation reduces the risk of local recurrence by about 50%. For decades, investigators have tried to identify subsets of DCIS for which the risk of local recurrence is low enough that a 50% reduction in relative risk results in such a small reduction in absolute risk that physicians and patients will not feel the benefit warrants the risk. Local recurrence after mastectomy for DCIS occurs in about 1% of cases, which is a level of risk that is not considered sufficient to warrant RT. The risk of local recurrence after lumpectomy without RT ranges as high as 20–30%. Factors that may impact this risk include the size of DCIS, histologic subtype, nuclear grade, resection margin status, and patient age. The first efforts in this were led by Dr. Melvin Silverstein at the Van Nuys Breast Clinic where he examined the rate of local recurrence in a large cohort of women treated with lumpectomy and no radiation. Based on the factors above, he proposed the “Van Nuys Index” to cases from whom the risk of local recurrence was low enough to omit radiation. However, subject randomized trials in the low-risk cohorts still showed local recurrence rates of 10–15%. Whether this is low enough to allow omission of RT is open to interpretation and RT is therefore still considered and discussed with most patients with DCIS. 

Omitting post-lumpectomy RT in patients with DCIS derives from the fact some studies show select patients have a low risk of ipsilateral breast tumor recurrence (IBTR) when treated with excision alone. In a study of 186 patients with DCIS who underwent lumpectomy alone, the 10-year disease free survival rates for low- and intermediate/high-risk DCIS were 94% and 83%, respectively [36]. A study of 215 patients treated with lumpectomy alone reported an 8-year recurrence rate of 0%, 21.5%, and 32.1% for low-, intermediate, and high-risk DCIS, respectively [37]. Conversely, some literature suggests that omitting RT is associated with a substantially higher rate of IBTR, and that although IBTR may be delayed, it is not prevented in seemingly low-risk DCIS. A non-randomized prospective study evaluated lumpectomy alone as a treatment for patients meeting either of two low-risk DCIS groups (low-/intermediate-grade ≤2.5cm DCIS or high-grade ≤1 cm DCIS) [38]. At a median follow-up of 12.3 years, IBTR were 14.4% and 24.6% for the low-/intermediate-grade and high-grade DCIS, respectively. The RTOG 9804 trial [33], which randomized 636 patients with low-risk DCIS to either RT or observation after surgery, showed that with a median follow-up of 7 years, local recurrence was lower after RT than observation (0.9% versus 6.7%; HR, 0.11; 95% CI, 0.03–0.47). At 15 years’ follow up, the local recurrence rates were reduced by 50% with RT compared to observation (0.9% versus 6.7%; HR, 0.11; 95% CI, 0.03–0.47). Two other prospective single arm trials, Eastern Cooperative Oncology Group (ECOG) 5194 [39] and Wong et al. [40], reported IBTR rates of 14.4% at 12 years and 15.6% at 10 years, respectively. Unfortunately, these studies were unable to identify the subset of patients with low-risk DCIS treated with lumpectomy alone who have local recurrence rates of <10% after long-term follow-up based on standard clinical and pathologic tumor features. Instead, a combination of clinico-pathological features and tumor biology may identify truly low-risk patients. For example, gene expression profiling (e.g.,: Oncotype DCIS) has been proposed. A meta-analysis using data from the ECOG E5194 and Ontario DCIS cohort aimed to refine the risk estimates provided by the Oncotype DCIS score by incorporating patient age and tumor size [41]. Patients in the low-risk group, defined as women ≥50 years of age with DCIS lesions ≤1 cm who had a low-risk DCIS score, had a 10-year local recurrence rate of 7.2%. DCISionRT, a biological signature comprised of biomarkers detected by immunohistochemistry and clinic-pathological features, may also aid in identifying women with low-risk DCIS with an 8% 10-year risk of IBTR [42].

De-escalation of DCIS requires caution. Synchronous invasive breast cancer remains a concern when patients are newly diagnosed with DCIS. Generally, patients are quoted a 20% risk of upgrade to invasive carcinoma upon surgical resection [43,44,45]. Even low-risk DCIS has a potentially high upgrade rate. For example, in a cohort of patients with DCIS, selected based on LORIS trial eligibility criteria, of those upgraded to invasive cancer, 31% were pT1b or larger and 5% were pN1 [46]. De-escalation will not be suitable for all patients, as one study suggested only 16.7% of women with DCIS meet low-risk criteria [46]. Metachronous invasive breast cancer may also have a worse effect on survival. Data from the National Surgical Adjuvant Breast and Bowel Project (NSABP) B-17 and B-24 trials show that women who developed an invasive ipsilateral recurrence after treatment for DCIS were at increased risk of dying of breast cancer (HR 7.06, 95% CI 4.14–12.03) [9]. Another study showed that in women with low-volume DCIS completely excised at the time of core biopsy, the 10-year rate of ipsilateral breast cancer in those who did not receive RT was 14.5% [47]. Given that women managed with observation are expected to develop higher rates of invasive cancers than those undergoing excision with or without RT, this approach may result in an increased mortality.

## 3. De-Escalation of Surgical Treatment for Invasive Breast Cancer

The local treatment of breast cancer has evolved considerably in the more than 125 years since William Halsted described the radical mastectomy in 1894 [48]. Building upon the work of other surgeons who had reported encouraging results with more extensive operations [49,50], Halsted outlined the en bloc removal of the breast, overlying skin, pectoralis major muscle, and regional lymph nodes. He also later emphasized removing the contents of the supraclavicular region [51]. Although such a radical operation for early breast cancer would be unrecognizable to modern surgeons, Halsted was guided by his sophisticated theory of orderly spread of breast cancer from the primary to regional lymphatics and thence to distant sites. The concept was that interruption of this orderly sequence by lymphadenectomy was advantageous. Indeed, the approach achieved impressive results compared to Halsted’s contemporaries. However, the results were primarily in achieving local control of advanced tumors. While local recurrence rates of prior case series ranged between 50 and 85%, Halsted recorded only a 6% recurrence rate [48]. However, as subsequent research would prove, extended surgery does not prevent metastatic spread or improve survival. While cancers still presented primarily as advanced local growth and in the absence of systemic therapies to address distant recurrence, the Halsted mastectomy was standard until the 1970s. 

Incremental advances in understanding breast anatomy and in treating many cancers before they were locally advanced led to the challenging of Halsted’s theories. Improved anatomical knowledge throughout the 20th century suggested that the deep fascia was not as rich in lymphatics as previously thought. Therefore, surgeons began experimenting with routine preservation of the pectoralis major. In 1948, Dr. David Patey [52] described the outcomes of 118 women with breast cancer treated with preservation of the pectoralis major, otherwise known as modified radical mastectomy (MRM) between 1930 and 1943. In this study, 46 women underwent MRM, while 45 women were treated with standard radical mastectomy. While the analysis was naïve by today’s standards, in that survival analysis was not yet widely utilized in medical research [53], 3-year survival was similar between the groups and longer-term follow-up confirmed the findings [54]. During these early efforts at de-escalation, the use of chest and regional nodal RT was evaluated and approaches familiar to modern surgeons were described. These included MRM with postoperative RT, partial mastectomy followed by radiotherapy, and the omission of axillary dissection in favour of axillary RT [55,56,57,58,59]. 

Despite these innovative clinical reports, substantial progress away from the Halsted radical mastectomy would not come until the 1970s with the initiation of several landmark clinical trials [60,61,62]. Watershed clinical trials that finally changed the paradigm of breast cancer treatment were started in the early 1970s by the National Surgical Adjuvant Breast Project under the leadership of Dr. Bernard Fisher. The first major concept testing in these trials was that extension of surgery to include the muscle and regional nodes did not improve survival, directly challenging Halsted’s hypothesis. This study, the NSABP B-04 trial, included women who were randomized between the Halsted radical mastectomy and simple mastectomy with preservation of the pectoralis major muscle. For those with clinically negative nodes, women were randomly assigned to have axillary dissection or no axillary dissection. This study demonstrated that radical surgery, and specifically regional node dissection, had no impact on survival, fundamentally changing the approach to breast cancer. 

Several groups began treating women with less than mastectomy and preserving the remaining breast (Table 2). At the National Cancer Institute in Milan, Italy, Veronesi et al. [61] randomized 701 women ≤ 70 years of age between 1973 and 1980 with breast cancer ≤2 cm in size and no clinically evident axillary disease to Halsted radical mastectomy or quadrantectomy plus RT. Quadrantectomy consisted of the radical removal of the affected quadrant of the breast, overlying skin, and fascia of the pectoralis major. Both groups underwent complete axillary dissection. The initial results of the trial were published in 1981 [61], demonstrating no significant differences in 5-year overall survival (radical mastectomy 90.1% versus quadrantectomy 89.6%; log-rank *p* = 0.88) or disease-free survival (radical mastectomy 83.0% versus quadrantectomy 84.0%; log-rank *p* = 0.54).

In the United States, the NSABP conducted a study with broader aims–the NSABP B-06. Fisher et al. [62] analyzed 1,843 women with breast cancers ≤4 cm in size (allowing non-matted clinically positive axillary disease) randomized into three groups (total mastectomy, segmental mastectomy, or segmental mastectomy plus RT). The extent of surgery in this trial was less radical than in the Veronesi study. During mastectomy, the pectoralis major was preserved, and a segmental mastectomy consisted of only enough tissue removal to ensure free margins from the tumour. The overlying skin and pectoralis fascia was generally not removed. The initial results published in 1985, with a mean follow-up of approximately 39 months, confirmed the findings of the earlier Italian study [62], showing no significant benefit to total mastectomy for either overall survival (*p* = 0.06) or disease-free survival (*p* = 0.9). Indeed, the addition of radiation to segmental mastectomy had no impact on survival. However, the NSABP-06 study demonstrated the importance of RT in the success of breast conserving therapy (BCT). The 5-year incidence of recurrent tumour in the ipsilateral breast was 7.7% in the segmental mastectomy group that received RT, and 27.9% in the segmental mastectomy alone group (*p* < 0.001). 

Both trials were updated with 20-year follow-ups in 2002, confirming the safety of breast conserving surgery when followed by RT [63,64]. Every subsequent study and meta-analysis confirmed the finding that when breast conserving therapy can be done appropriately, there is no overall outcome advantage to mastectomy. Further, with improved surgery and radiation techniques, and with what is now standard adjuvant systemic therapy, the local recurrence advantage of mastectomy may be disappearing, with many reports showing local recurrence rates with breast conserving surgery and radiation under 5%, which is the same rate as with mastectomy with negative nodes. Questions remained regarding the most appropriate procedure, and it was challenging to compare the trials, given differing inclusion criteria. Ultimately, the lumpectomy technique in NSABP B-06 has become the dominant method of BCT. It is less disfiguring than quadrantectomy, and sufficient evidence has accumulated that shows no benefit to margins wider than “no ink on tumour” after BCT [25,65,66,67,68]. Numerous localization techniques for non-palpable tumours have further enhanced the surgical technique [69]. However, reoperation rates after BCT for positive or close margins is an important balancing factor [70].

While BCT has become the preferred treatment for breast cancer, there remain relative and absolute contraindications to BCT [71]. Historically, these were scenarios that would result in potentially high rates of a subsequent ipsilateral recurrence of new cancer or poor cosmetic outcomes. This includes inability to obtain negative margins, the presence of more than one cancer in the breast, or having a large cancer in relation to the size of the breast. Additionally, the inability to receive RT may be a contraindication, which includes active collagen vascular disease. Historically, women with breast cancer during pregnancy, especially the first trimester, were thought to require mastectomy. However, with current use of neoadjuvant therapy, and potentially short delays in delivering RT until after delivery, some women with breast cancer in pregnancy can be treated with BCT. Other relative contraindications included previous chest wall RT, inflammatory breast cancer, an extensive area of DCIS, or inability to obtain negative margins, and hereditary breast cancer. 

Advances continue in the surgical treatment of invasive breast cancer. More widespread use of neoadjuvant chemotherapy has safely increased the number of women eligible for BCT, who would previously have received mastectomy [72,73]. Some data have also shown that BCT in the setting of multicentric disease can have acceptable results. The Alliance Z11102 trial is a single-arm trial investigating the feasibility of BCT in patients with multiple ipsilateral breast cancer (tumours separated by 2 cm or greater of normal tissue) [74]. Initial results among 198 women demonstrated low rates of ultimate conversion to mastectomy for positive margins (7.1%), favorable cosmetic outcomes, and 74.6% of BCT patients underwent a single operation [75,76]. The trial’s primary outcome of 5-year local recurrence is expected to mature soon and should provide an important update on this historical barrier to BCT. Another evolving standard is the fact that mastectomy may not be necessary for women with hereditary breast cancer in terms of long-term survival [77], though many women choose this because of the very high risk of a subsequent new breast cancer in either breast. Finally, the development of oncoplastic surgery has allowed patients who may otherwise need mastectomy with/without reconstruction to pursue breast conservation while not compromising cancer outcomes [78].

## 4. De-Escalation of Surgery after Neoadjuvant Systemic Therapy

Systemic therapy for breast cancer (e.g., chemotherapy, immunotherapy, or endocrine therapy) aims to treat not only the primary tumor, but also microscopic metastatic disease, with the aim of reducing the risk of local and distant recurrence. The Veronesi and NSABP trials of BCT were conducted in the 1970s when most women did not receive any adjuvant systemic therapy. The efficacy of these treatments, especially in node-negative cancer, were not proven until the late 1980s. Such treatments have a significant impact on the risk of local recurrence, with rates well below 5% now expected with BCT. More recent efforts are also aiming to develop the use of neoadjuvant endocrine therapy for locally advanced hormone-receptor positive HER2 negative breast cancer with the goal of down-staging tumors and facilitating breast conservation [79].

Equally important has been the increasingly widespread use of neoadjuvant systemic therapy for breast cancer. This has been shown to down-stage breast cancer to allow for less invasive surgery [80], namely BCT instead of a mastectomy, or a sentinel lymph node biopsy instead of an axillary lymph node dissection. For example, neoadjuvant systemic therapy has allowed patients who become node-negative to undergo sentinel lymph node biopsy to evaluate nodal pathological complete response [81]. However, even with less invasive surgery, patients are at risk of morbidity, including chronic pain and lymphedema [82,83,84,85]. Therefore, efforts to further de-escalate axillary lymph node surgery are becoming necessary. An example is the Alliance A011202 trial, in which patients with residual nodal disease after neoadjuvant systemic therapy are being randomized to axillary lymph node dissection compared to axillary radiation therapy (NCT01901094).

In some patients who undergo neoadjuvant systemic therapy followed by BCT, the surgical specimen displays no evidence of residual disease on final pathology – a complete pathological response (pCR). While the impact on survival of response to systemic therapy is the same whether it is administered before or after surgery, having a pCR identifies those who have benefited from systemic therapy in that they have decreased disease recurrence and increased overall survival [86,87,88,89,90,91]. If pCR truly reflects complete resolution of the tumor after systemic therapy, then perhaps satisfactory oncologic outcomes can be obtained while avoiding surgery entirely. One major obstacle is that the current gold standard for determining pCR is analysis of a surgical specimen [92]. Therefore, it is necessary to establish the validity of non-surgical techniques for identifying pCR.

To this end, there have been several trials exploring the efficacy of biopsy, imaging, and/or biomarkers to determine, without surgery, who has had a pCR [93,94,95]. A common goal of these trials is to minimize the false negative rate (FNR), the proportion of patients who are identified to have pCR prior to surgery, but who ultimately have residual disease on final pathology. The RESPONDER trial [95,96] attempted to prove that an image-guided vacuum-assisted biopsy (VAB) of the original site of the breast cancer could diagnose pCR with an FNR of less than 10%. This primary endpoint was not achieved with VAB alone, demonstrating an FNR of 17.8% (95% CI, 12.8–23.7%). However, a secondary exploratory analysis of the data determined that the combination of breast imaging and VAB reduced the FNR to 6.2% (95% CI, 3.4–10.5%). A subsequent analysis [97] showed that multivariate algorithms, using patient and tumor factors in addition to VAB results, was able to predict pCR with an FNR of only 1.2%. If this algorithmic strategy can be further validated, it may become feasible to routinely detect patients with pCR without a surgical specimen.

Once pCR is proven in a patient who has received neoadjuvant systemic therapy, it is necessary to establish the efficacy of further non-surgical treatment. Even in patients who undergo surgery, pCR is not a guarantee of disease-free survival, although it is a strong predictor. In a 2020 meta-analysis by Spring et al. of 27,895 patients across 52 publications, 5-year OS for patients who had pCR was 94% [95% prediction interval (PI), 90–96%], compared to 75% (95% PI: 65–82%) for those who did not [90]. The highest association between pCR and survival was seen with triple negative breast cancer (HR 0.20, 95% PI, 0.07–0.41), but this was also the subtype of cancer with the lowest 5-year OS, 84% (95% PI, 60–97%) in those with pCR and 47% (95% PI, 13–77%) in those without pCR. The 5-year mortality, even in patients with pCR, is non-negligible; therefore, omitting surgery may worsen the mortality rate. Interestingly, this same meta-analysis showed no difference in OS or event-free survival in patients with pCR receiving adjuvant chemotherapy, compared to those who did not. While there is a growing body of research on additional systemic therapy for patients who do not attain pCR, that is beyond the scope of this review.

A primary treatment modality for post-neoadjuvant systemic therapy patients with pCR who do not undergo surgery may need to include local control with RT. Currently, a clinical trial (NCT02945579) aims to determine local disease recurrence in patients who undergo neoadjuvant systemic therapy followed by RT without surgery [98]. RT does also offer its own level of morbidity, and there is a separate arm of this trial that aims to study disease recurrence in patients who solely undergo BCT after neoadjuvant systemic therapy without RT. While this is a relatively small trial, only aiming to enroll 50 patients, it is foreseeable that if there are comparable outcomes in patients receiving BCT or RT after neoadjuvant systemic therapy compared to standard BCT, there may be room for larger trials in the future. If there is no survival benefit for either surgery or RT after neoadjuvant therapy, perhaps there will be room to discuss neoadjuvant systemic therapy as definitive therapy when pCR is attained, with the avoidance of surgery and RT altogether.

If patients are to undergo a surgery-sparing regimen after neoadjuvant systemic therapy, certain aspects require further investigation. It will be important to maintain adequate surveillance to monitor for recurrent disease. The FNR for determining pCR without surgery will never be 0%, and there will be patients in whom residual disease is missed. Surveillance strategies will require close monitoring. Ideally, patients failing nonoperative management will be identified at an early enough stage that they can undergo salvage therapy with surgery and/or additional systemic therapy. Further studies will be needed to clarify surveillance strategies once the feasibility of a no-surgery treatment course is established. Additionally, it will also be necessary to identify the patient population that benefits the most from a no-surgery approach. Most likely, it will involve those patients who have small tumors with limited or absent nodal involvement, and with a hormone receptor status that is amenable to effective neoadjuvant therapy. It will also require an open and honest conversation with patients, who will need to make a personal choice. 

## 5. Conclusions

De-escalation of breast cancer treatment is a key area of investigation that will continue to remain a priority. This process requires reproducible identification of patients with low-risk breast cancer that will not have their oncologic outcomes affected. Future efforts should continue to challenge breast cancer treatment modalities and provide insight into how this affects patient reported outcomes 

## Figures and Tables

**Table 1 cancers-14-04545-t001:** Summary of international clinical trials evaluating active image-based surveillance for patients with low-risk DCIS.

	COMET [28]	LORD [27]	LORETTA [30]	LORIS [29]
**Coordinating country**	United States of America	Netherlands	Japan	United Kingdom
**Phase**	III	III	III	III
**Study design**	Randomized controlled trial	Patient preference	Single arm	Randomized controlled trial
**Eligible age range (years old)**	≥40	≥45	≥40, ≥75	≥46
**Year of study activation**	2017	2017	2017	2014
**Target accrual**	1200	1240	340	932 (closed 2020)
**DCIS maximum size**	Any size	Any size	2.5 cm	-
**Nuclear grade**	1 or 2	1	1 or 2	1 or 2
**Comedo necrosis**	Yes	No	No	No
**Estrogen receptor**	Positive	-	Positive	-
**HER2**	Negative (if tested)	-	Negative	-
**Endocrine therapy**	Permitted	Not permitted	Permitted	Permitted
**Primary outcome**	2, 5, and 7 years	10 years	5 and 10 years	10 years

**Table 2 cancers-14-04545-t002:** Summary of prospective randomized clinical trials comparing breast conserving therapy with mastectomy.

Clinical Trial	N	Tumor Size (cm)	Margin	Interval Reported (Years)	Local Recurrence Rate (%)		Overall Survival Rate (%)	
					Breast Conserving Therapy	Mastectomy	Breast Conserving Therapy	Mastectomy
**NSABP**	1851	4	Tumor free	20	14	10	46	47
**EORTC**	868	5	1 cm	20	20	12	65	66
**Danish**	793	Any	Grossly free	20	-	-	58	51
**Milan**	701	2	-	20	9	2	42	41
**NCI**	247	5	Grossly free	25	22	6	59	58
**IGR**	179	2	2 cm	15	9	14	73	65

Abbreviations: EORTC, European Organization for Research and Treatment of Cancer; IGR, Institut Gustave-Roussy; NCI, National Cancer Institute; NSABP, National Surgical Adjuvant Breast and Bowel Project.

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
