# Peer review of "De-Escalating the Management of In Situ and Invasive Breast Cancer"

_cancers, 2022, doi:10.3390/cancers14194545_

Round 1
Reviewer 1 Report
Comment to the Authors
The manuscript is overall well written and comprehensive, consider adding a small paragraph regarding the most recent updates on nodal management and de-escalation strategies as per above.
See below few editorial changes
Abstract:
The abstract gives an appropriate summary of the manuscript
- Line 35 - If adding abbreviation for DCIS also add abbreviation of “neoadjuvant systemic therapy (NST).
Introduction
- Line 53 - remove double space after the “.”
- Line 54 - I would replace the word “work” with “research”
- Line 60 - add abbreviation of “neoadjuvant systemic therapy (NST)
De-escalation of DCIS treatment
- Line 71 - the sentence “however, local recurrence may occur with breast conservation occurs as DCIS and invasive carcinoma in equal proportions” does not read well, consider changing to “however, local recurrence after breast conservation occurs for DCIS and invasive carcinoma in equal proportions”
- Line 82 – “stating that 25-50% of untreated DCIS will progress into invasive cancer” is not insignificant, among the references cited here, according to the most updated one [Ryser 2019] the 10-year cumulative incidence of ipsilateral invasive cancer is 10.5%. Please review the sentence and the reported %
- Line 88 – Considering this is a debated topic, literature citation/s should be added after this sentence.
- Line 103 – remove “continental” before Europe
- Line 106 – add “is” between trials and development
- Line 147 - remove double space after the “.”
De-escalation of local treatment for invasive breast cancer
- Line 290 – “ductal carcinoma in situ” with DCIS
- Consider the historical review from radical mastectomy to BCS I would consider adding a sentence on the most modern oncoplastic approach to breast conservation at the end of the paragraph [Silverstein 2019 on extreme oncoplasty could be a good reference here]
De-escalation of surgery after neoadjuvant systemic
Line 307 – the concept of neoadjuvant endocrine therapy is here anticipated, however is not expanded further on the paragraph. Consider adding the recent published advantages on NAET would improve the comprehensiveness of the paragraph [Barchiesi 2020 offers a comprehensive review of the latest data]
- Line 314 “Neoadjuvant systemic therapy” has been already abbreviated, replace with NST
- Line 318 – the advantage de-escalating nodal management after NST is here detailed. A small paragraph should be added to explain the advantage of NST on both local and locoregional downgrade. This should be further expanded with ACOSOG Z1071 results and current Alliance A011202 trials.
- Line 324 – remove double space between “have” and “decreased”
Author Response
- Line 35 - If adding abbreviation for DCIS also add abbreviation of “neoadjuvant systemic therapy (NST)
Response: We thank the reviewer for their suggestion. We have noted that the abbreviation of NST is not standard/common. We have removed it from the manuscript and therefore would not be necessary in the abstract.
- Line 53 - remove double space after the “.”
Response: We thank the reviewer for noticing this. The extra space has been removed.
- Line 54 - I would replace the word “work” with “research”
Response: The word “work: has been replaced for “research”.
- Line 60 - add abbreviation of “neoadjuvant systemic therapy (NST)
Response: We thank the reviewer for their suggestion. We have noted that the abbreviation of NST is not standard/common. We have removed it from the manuscript and therefore would not be necessary in the abstract.
- Line 71 - the sentence “however, local recurrence may occur with breast conservation occurs as DCIS and invasive carcinoma in equal proportions” does not read well, consider changing to “however, local recurrence after breast conservation occurs for DCIS and invasive carcinoma in equal proportions”
Response: We thank the reviewer for this suggestion. The sentence has been changed to “However, local recurrence after breast conservation occurs for DCIS and invasive carcinoma in equal proportions”
- Line 82 – “stating that 25-50% of untreated DCIS will progress into invasive cancer” is not insignificant, among the references cited here, according to the most updated one [Ryser 2019] the 10-year cumulative incidence of ipsilateral invasive cancer is 10.5%. Please review the sentence and the reported %
Response: We thank the reviewer for this suggestion. We have modified the sentence and now reads as follows: “Left untreated, 10.5% to 50% of patients with DCIS may progress to invasive carcinoma”.
- Line 88 – Considering this is a debated topic, literature citation/s should be added after this sentence.
Response: We thank the reviewer for this suggestion. Four citations have been added to support this statement.
- Line 103 – remove “continental” before Europe
Response: We thank the reviewer for this suggestion. The word “continental” has been removed.
- Line 106 – add “is” between trials and development
Response: We thank the reviewer for this suggestion. The word “is” has been added.
- Line 147 - remove double space after the “.”
Response: We thank the reviewer for noticing this. The extra space has been removed.
- Line 290 – “ductal carcinoma in situ” with DCIS
Response: We thank the reviewer for this. The acronym DCIS has been used instead.
- Consider the historical review from radical mastectomy to BCS I would consider adding a sentence on the most modern oncoplastic approach to breast conservation at the end of the paragraph [Silverstein 2019 on extreme oncoplasty could be a good reference here]
Response: We thank the reviewer for their suggestion. The topic of oncoplastic surgery has been added at the end of the paragraph along with the suggested reference. The new sentence is the following “Finally, the development of oncoplastic surgery has allowed patients who may otherwise need mastectomy with/without reconstruction to pursue breast conservation while not compromising cancer outcomes”.
- Line 307 – the concept of neoadjuvant endocrine therapy is here anticipated, however is not expanded further on the paragraph. Consider adding the recent published advantages on NAET would improve the comprehensiveness of the paragraph [Barchiesi 2020 offers a comprehensive review of the latest data]
Response: We thank the reviewer for their suggestion. The suggested reference has been added along with the following sentence: “More recent efforts are also aiming to develop the use of neoadjuvant endocrine therapy for locally advanced hormone-receptor positive HER2 negative breast cancer with the goal of down-staging tumors and facilitating breast conservation.”
- Line 314 “Neoadjuvant systemic therapy” has been already abbreviated, replace with NST
Response: We thank the reviewer for their suggestion. We have noted that the abbreviation of NST is not standard/common. We have removed it from the manuscript and therefore would not be necessary in the abstract.
- Line 318 – the advantage de-escalating nodal management after NST is here detailed. A small paragraph should be added to explain the advantage of NST on both local and locoregional downgrade. This should be further expanded with ACOSOG Z1071 results and current Alliance A011202 trials.
Response: We thank the reviewer for their suggestion. We have included the following sentences, with the corresponding references to this paragraph to expand on de-escalating nodal management: “For example, neoadjuvant systemic therapy has allowed patients who become node-negative to undergo sentinel lymph node biopsy to evaluate nodal pathological complete response . However, even with less invasive surgery patients are at risk of morbidity including chronic pain and lymphedema [75-76], poor cosmesis [77], and lymphedema [78]. Therefore, efforts to further de-escalate axillary lymph node surgery are becoming necessary. An example is the Alliance A011202 trial, in which patients with residual nodal disease after neoadjuvant systemic therapy are being randomized to axillary lymph node dissection compared to axillary radiation therapy (NCT01901094).”
- Line 324 – remove double space between “have” and “decreased”
Response: We thank the reviewer for noticing this. The extra space has been removed.
Reviewer 2 Report
This paper provides a thorough yet concise review of deescalated treatments for DCIS (observation vs surgery), breast cancer (smaller surgeries for similar outcomes), and avoiding surgery after neo adjuvant chemotherapy. The paper is easy to read and contains all the major trials and a multitude of citations providing a comprehensive summary of these three areas. It will make a good reference and starting point for anyone interested in deescalation of treatment. The only cons are a few minor grammatical errors and that some of the trials for this area of study have not produced results yet which cannot help guide our management. Overall I would recommend publishing this article for the value it provides as an in depth over view.Overall nice review of the history of treatment and de escalation attempts. Well written and easy to read.
Author Response
We thank the reviewer for their time assessing out manuscript. We have reviewed the whole manuscript for grammar.